# Voltammetric Electronic Tongue for the Simultaneous Determination of Three Benzodiazepines

**DOI:** 10.3390/s19225002

**Published:** 2019-11-16

**Authors:** Anna Herrera-Chacón, Farzad Torabi, Farnoush Faridbod, Jahan B. Ghasemi, Andreu González-Calabuig, Manel del Valle

**Affiliations:** 1Sensors and Biosensors Group, Department of Chemistry Universitat Autònoma de Barcelona, Edifici Cn, 08193 Bellaterra, Spain; anna.herrerachacon@gmail.com (A.H.-C.); farzad.torabi@ut.ac.ir (F.T.); andreugc27@gmail.com (A.G.-C.); 2Center of Excellence in Electrochemistry, Faculty of Chemistry, University of Tehran, Tehran 1417466191, Iran; faridbodf@khayam.ut.ac.ir; 3School of Chemistry, College of Science, University of Tehran, Tehran 1417466191, Iran; jahan.ghasemi@ut.ac.ir

**Keywords:** electronic tongue, nanoparticle modifiers, diazepam, lorazepam, flunitrazepam, artificial neural networks

## Abstract

The presented manuscript reports the simultaneous detection of a ternary mixture of the benzodiazepines diazepam, lorazepam, and flunitrazepam using an array of voltammetric sensors and the electronic tongue principle. The electrodes used in the array were selected from a set of differently modified graphite epoxy composite electrodes; specifically, six electrodes were used incorporating metallic nanoparticles of Cu and Pt, oxide nanoparticles of CuO and WO_3_, plus pristine electrodes of epoxy-graphite and metallic Pt disk. Cyclic voltammetry was the technique used to obtain the voltammetric responses. Multivariate examination using Principal Component Analysis (PCA) justified the choice of sensors in order to get the proper discrimination of the benzodiazepines. Next, a quantitative model to predict the concentrations of mixtures of the three benzodiazepines was built employing the set of voltammograms, and was first processed with the Discrete Wavelet Transform, which fed an artificial neural network response model. The developed model successfully predicted the concentration of the three compounds with a normalized root mean square error (NRMSE) of 0.034 and 0.106 for the training and test subsets, respectively, and coefficient of correlation R ≥ 0.938 in the predicted vs. expected concentrations comparison graph.

## 1. Introduction

Benzodiazepines (BZs) are a family of drugs massively used worldwide since the 1960s, when they were first introduced. BZs are frequently prescribed as anxiolytics, anticonvulsants, muscle relaxants, and as treatment for alcohol and drug abuse [1]. At the same time, they are also used in clinical anesthesia due to their sedative and relaxant properties in a variety of procedures. The clinical use of these compounds is accomplished through their effects on the central nervous system (CNS) and specifically through the modulation of the inhibitory neurotransmitter gamma-aminobutyric acid (GABA) receptor [2]. BZs enhance GABA_A_ receptor affinity producing sedative, tranquilizing, and sleep-inducing effects. BZs supposedly present low toxicity when administered at clinical doses but may be potentially dangerous when used at high doses. Even though they are well-known drugs, they are difficult to identify in the body after their use. Therefore, the identification of BZs in blood [3], urine [4], hair samples [5], or in post-mortem studies may be interesting in terms of criminology and forensic studies [6].

A plethora of analytical methods to determine BZs can be found in the literature such as spectrophotometry [7], capillary electrophoresis [8], HPLC/MS [9], LC–MS [10], and immunoassays [11,12]. However, these techniques require high-cost instrumentation, are slow, require highly trained personnel, and cannot be implemented to perform on-line measurements. Among less employed techniques, electrochemical methods stand out, specifically, voltammetric methods. Voltammetric methods have several advantages over classical methods, including good sensitivity, low detection limits, relative simplicity, portability, and low-cost instrumentation compared to other techniques. They also have enough versatility to be implemented with portable instrumentation. The study of BZs by electrochemical techniques is centered on the determination and the quantification of a single BZ at a time. As an example of individual BZ studies we mention determining clonazepam in beverages and serum samples using screen printed electrodes [13] or using polyaniline/graphene oxide nanocomposites [14]; detecting diazepam by dendritic silver nanostructures supported by graphene [15], studying its redox behavior in drinks by adsorptive stripping voltammetry [16], or by the use of fullerene-functionalized carbon nanotubes [17]; determining lorazepam by polypyrrole@sol-gel@gold nanoparticles pencil graphite electrode [18]; quantifying olanzapine using quantum dots onto modified multiwalled carbon nanotubes gold electrode [19]; and, lastly, sensing flunitrazepam in screen-printed electrodes [20], or in alcoholic and soft drinks by screen-printed drop-volume cells [21]. The determination of several BZs can be found in the recent literature as the determination of diazepam and oxazepam in biological fluids using modified carbon paste electrodes [22], or with the use of a multiwall carbon nanotube-ionic liquid paste electrode [23]; using a boron-doped diamond electrode to determine bromazepam and alprazolam [24]; an attempt to determine olanzapine and risperidone modifying gold electrodes with carbon nanotubes [25]; and using a modified bentonite sonogel carbon electrode to determine diazepam chlordiazepoxide determined in urine [26]. Furthermore, the identification or determination of different BZs by single sensors such as monitoring five BZs using a polymeric interfaces electrode modified in layer by layer manner [27], applying square-wave voltammetric technique using a hanging mercury drop electrode [28], using silver nanoparticles-carbon dots ink modified electrodes [29], or employing a modified silver solid amalgam electrode can be found in the literature [30]. However, none of these works accomplishes the simultaneous determination of BZs, since they do it individually. Consequently, the simultaneous voltammetric determination of multiple BZs is still a challenge, mainly due the high overlapping of redox peaks when multiple BZs are present in the same solution. As the specific problem concerns finding a sensor-based analytical method that is able to determine and quantify different BZs simultaneously, the use of a sensor array becomes an interesting option. For this reason, this work aims to use an electronic tongue approach to solve a ternary mixture of three BZs: diazepam, flunitrazepam, and lorazepam, three regulated substances with equivalent functional base, and varying substituents (Figure 1).

Electronic tongues (ETs) have been in use since mid-1990s to overcome limitations of single sensor methodologies, spreading the use of sensor arrays and chemometric tools. An ET consists of a multisensor system formed by low-selective sensors, and uses advanced mathematical procedures for signal processing based on pattern recognition and/or multivariate analysis, among these Artificial Neural Network (ANNs) or Principal Component Analysis (PCA) [31]. These mathematical tools are widely used to correct interference and/or matrix effects in analytical procedures. In recent published literature, electronic tongues have been used to differentiate and quantify analogue species that, thanks to the gathering of multiple sensor data, made possible the successful resolution of situations for which traditional techniques function with difficulty. ETs can be classified or divided depending on the techniques employed, including potentiometric [32,33], amperometric [34], impedance based [35,36,37] and voltammetric [38]. Some examples in recent literature have adopted voltammetric ET principles to discriminate red wines [39], to asses wine sensory descriptors [40], to detect and quantify nitro and peroxide explosives [41], and to distinguish aminothiols [42]. An important breakthrough in this field has been the generic approach employing molecularly imprinted polymers to form (bio)sensor arrays [43].

In the present paper, an efficient ET approach is suggested for the simultaneous determination of ternary mixtures of BZs comprising diazepam, flunitrazepam, and lorazepam using chemometric tools coupled with an electrochemical detection system. Multidimensional voltammetric information is obtained through the use of an array of different modified electrodes with a carbon base. For this purpose voltammogram data were compressed using the Discrete Wavelet Transform (DFT), and the resulting coefficients were used as input for the Artificial Neural Network (ANN) model as a single vector formed by all the pre-treated signals. The resolution of any signal overlapping and quantification of the individual species considered is achieved thanks to the extraction of the samples’ fingerprints along with the capabilities of the ANN predictive model. Satisfactory models for identification and resolution of their mixtures were thus developed. Figure 2 shows schematically the voltammetric ET strategy followed. The approach comprises succeeding stages: (A) an experimental design is used to define the standards needed to build the response model; (B) these are measured using a voltammetric sensor array; (C) the data obtained is compressed and fingerprints extracted; (D) the model is trained with the data extracted from the measurements; and (E) the model is validated with an independent set of samples.

## 2. Materials and Methods.

### 2.1. Chemicals and Reagents

All chemicals used in this investigation were of analytical reagent grade. Water was purified by a Milli-Q Ultra Pure Water System (Millipore, Bedford, MA, USA). Diazepam, flunitrazepam, and lorazepam were purchased from Sigma-Aldrich (Madrid, Spain). All the samples were diluted in Britton-Robinson Buffer, pH 10 (0.04 M H_3_BO_3_, 0.04 M H_3_PO_4_, 0.04 M CH_3_COOH, and 0.1 M NaOH). Modifiers employed in the preparation of electrodes were purchased from Sigma-Aldrich: platinum nanopowder, 50 nm particle size (Ref. 6845453); copper (II) oxide, 50 nm particle size (Ref. 544868); copper nanopowder, 40–60 nm particle size (Ref. 774111); and tungsten (VI) oxide, 100 nm particle size (Ref. 550086).

### 2.2. Preparation of the Modified Electrodes

The main electrodes employed in this work are graphite epoxy composite electrodes modified in different variants. In order to build the electrodes a copper disk was soldered to an electrical connector. The connector was then fitted into a 6 mm diameter PVC tube, which provided the main body of the electrode. Then a conductive paste was prepared using 50-µm particle size graphite powder (BDH laboratory Supplies, Poole, UK) and Epotek H77 resin and hardener (both from Epoxy Technology, Billerica, MA, USA), which were deposited, filling the cavity in the void body. The electrodes were cured in the oven for 72 h and then repeatedly polished with sandpaper of decreasing grains prior to their readiness for measuring (see Figure 3).

Modified electrodes follow the same building process but instead of using a paste of graphite and epoxy resin, they incorporate a 5% of modifiers acquired from commercial source. In this work, the modifiers employed were metallic nanoparticles of Cu and Pt, oxide nanoparticles of CuO and WO3, a Pt disk electrode, and a graphite epoxy electrode (GEC) as a bare electrode.

The use of such graphite epoxy electrodes modified with nanotechnological components has been reported in our laboratory for its use in the detection of Brett character in wine [44], to asses wine sensory descriptors from a sensory panel [40], to analyze amino acid mixtures [45], to evaluate the antioxidant capacity of red wines [46], to discriminate beers [47], to analysis cava beverages [48], and for the comparison of different electronic tongues for pharmaceutical analysis [49].

As mentioned before, noble metal disc electrodes were also employed in the array. These electrodes were constructed soldering a Pt wire (99.95% purity, diameter 1 mm, supplied by Goodfellow, Cambridge, UK) to an electrical connector and then encasing the connector in the PVC tube (Servei Estació, Barcelona, Spain). The wire was finally coated in epoxy resin, cured and polished to let exposed only the wire cross section.

### 2.3. Electronic Tongue

The voltammetric ET was formed by an integrated array of six modified graphite epoxy electrodes as working electrodes, described in the previous section. The electrochemical cell employed in the measurements was formed by 6 working electrodes, an Ag/AgCl reference electrode, and a platinum electrode as counter electrode (physical area 0.47 cm^2^).

Electrochemical measurements were performed at room temperature (25 °C), using a 6-channel AUTOLAB PGSTAT20 (Metrohm Autolab BV, Utrecht, The Netherlands) controlled with the GPES 4.7 Multichannel software package.

A complete voltammogram was recorded for each sample by cycling the potential between −1.5 V and +1.5 V vs. Ag/AgCl with a step potential of 9 mV and a scan rate of 100 mV·s^−1^. Acquired voltammogram for data processing was the fourth (in a series of 4) to improve reproducibility of measurements. A conditioning potential of +1.2 V was applied during 60 s in a pH 10 solution after each measurement to electrochemically clean the electrochemical surfaces, thereby preventing fouling and drifting effects.

### 2.4. Model Design and Sample Preparation

In order to build an acceptable model with good generalization capabilities a considerable number of data points are usually needed for the training of the model. As stated in the European Pharmacopoeia guidelines for multivariate calibrations [50], the size of the dataset needed for building the calibration is dependent on interfering properties and the number of analytes that the model is expected to handle. Thus, this leads to two options: increase the size of data set to overcome the variability of the measurements [51], or employ a statistically defined data set [41].

In this case, the second option was preferred. The model employed was based on a factorial design [52] with 3 levels and 3 factors (27 samples). The final model was obtained after the factorial was tilted 45° in each axis for a better final representation. The training subset comprised 75% of the total samples.

Once the model was trained, an external test subset of 9 samples (25%) was employed to validate the response model. The concentrations of each sample of the test subset were generated randomly within the limits of the experimental domain (0–30 ppm) (Figure 4).

Both sample subsets were prepared in the same way and the samples were prepared in a Britton–Robinson buffer, pH = 10.0. In order to minimize the degradation of the chemical species in solution and reduce a possible source of data variability, fresh stock solutions of diazepam, flunitrazepam, and lorazepam were prepared on the same day the measurements were made.

### 2.5. Data Processing

Statistical treatment and data analysis were performed using routines written by the authors using MATLAB R2017a (MathWorks, Natick, MA, USA), together with its Neural Network and Wavelet toolboxes. Sigmaplot (Systat Software Inc., San Jose, CA, USA) was used to graphically represent and analyze the results.

## 3. Results

### 3.1. Voltammetric Array Response

Electrochemical response of BZs is largely dictated by its common core element, i.e., the 4,5-azomethine group. The electrochemical behavior has been explained by the 2e^−^, 2H^+^ reduction of the 4,5-azomethine group at the ring to give the corresponding dihydro species (see Figure 5), plus extra reduction of additionally present active moieties, e.g., the nitro group in the case of flunitrazepam.

In Figure 6 we can see an example of the different voltammograms obtained for 25 ppm of diazepam, flunitrazepam, and lorazepam in a Britton–Robinson buffer at pH 10.0 (conditions according to similar works in the literature [22]), for: (A) bare epoxy-graphite composite electrode, (B) CuO nanoparticles modified graphite electrode, (C) Cu nanoparticles modified graphite electrode, (D) Pt nanoparticles modified graphite electrode, (E) WO_3_ nanoparticles modified graphite electrode, and (F) Pt metal electrode. As can be seen in Figure 6, each sensor provides a singular voltammetric response. The employed sensors presented different sensitivities towards the different BZs, displaying a complementary redox profile. This being one of the necessary conditions for obtaining a properly functioning ET, obtaining differentiated signals for each considered substance with each sensor.

In order to mathematically assess whether the selected sensor array was able to provide the necessary signal complementarity when the different BZs are measured, a Principal Component Analysis (PCA) treatment was performed on a series of measurements using the three substances studied. Specifically, this was treated with separate voltammograms for each electrode and each sample employed in this study. With this multivariate visualization, each sensor appears differentiated (showing absence of collinearity) and even the different chemical compounds group together, suggesting the proper clustering for the intended application (see Figure 7). Moreover, and as can be seen in the score plot, the sensor’s scores are located in different quadrants of the plot, demonstrating complementarity of the sensors employed. These appear also away from the 0,0 origin, which suggests they provide useful information for the discrimination of samples.

### 3.2. Quantitative Analysis: Artificial Neural Network Modeling

An ANN response model was built from the set of compressed voltammograms in order to predict the three BZs concentrations in a given sample. For this a training subset of standards (n = 27) was used as defined by a tilted factorial design. Next, the prediction and generalization capabilities of the model were tested by employing a subset of unrelated and unknown samples to our model, i.e., the test subset (n = 9), defined by random concentration values of the three BZs.

As commented, given the complexity of the electrochemical information obtained from each sample, it is necessary to perform a pre-processing compression step. This step is necessary in order to avoid over-fitting of the developed models and to reduce the computation time spent during the training of the different models [31]. In this particular case, to reduce the data complexity obtained from the samples, the voltammetric data, the Discrete Wavelet Transform (DWT) was employed to reduce the 4512 current values obtained per sample (i.e., the six voltammograms) to a set of 144 wavelet coefficients per sample, in this case the approximation coefficients. The mother wavelet used was Daubechies 3, and the compression level used was 3. Values of compression achieved in the preprocessing step for the different compression levels assayed were 88.9% for the first, 94.1% for the second, and 96.7% for the third variant, the latter of which was chosen.

The obtained model was composed of 144 neurons in the input layer (24 per sensor), five neurons in the hidden layer, and three neurons in the output layer. The transfer functions employed in the hidden layer and output layer were satlins and tansig, respectively. These configuration details were chosen after exhaustive evaluation of roughly 200 different possibilities for the ANN, originating in a test of a number between two and nine neurons in the hidden layers, and the use of the five transfer functions hardlims, satlins, purelin (linearly shaped), and satlins and tansig (S-shaped) both in the hidden and output layers. Choice was performed looking for the minimum normalized Root Mean Squared Error as the measure used to quantify the fitting degree.

Performance of the trained ET is summarized in Figure 8. Comparison graphs of obtained vs. expected BZs concentrations were built in order to visualize the prediction ability of the ANN model. Moreover their best linear regression line was calculated (separately for the training and the testing subsets) for the three different BZs. Ideal values for the comparison lines should be 1.0 for the slope, 0.0 the intercept, and 1.0 for the correlation coefficient, respectively. The figure shows a satisfactory fit for the training data and also the correct trend for the external test subset, with perhaps a somewhat higher dispersion for the latter. This is the usual result when the proper modelling has been attained—almost perfect equivalence for the training subset, given the model is built from this data, and a correct trend for the external test subset, perhaps with less accuracy. However it must be said these data were not involved in the development of the model. Correlation coefficients indicated on the graph are practically 1.0 for the training subset and between 0.938–0.961 for the test subset, values also close to ideality.

The detailed results of regression lines for each compound and both train and test sub-sets are presented in Table 1. Acceptable values close to the ideal values were obtained for all the compounds with better values for the training set than for the testing set as it is usually obtained. Additionally, the table also includes NRMSE values as a measure of the goodness of fit with figures close to zero for both training and testing sets.

Finally, to give a contrast to the results herein, Table 2 was prepared for the examination and comparison of the performance of the present work and those of related papers in the literature. As can be seen from Table 2 linear ranges and limits of detections are similar to those of works in the literature but the presented work has the clear advantage of simultaneously quantifying three BZs species.

## 4. Conclusions

The proposed approach reports the use of a voltammetric sensor array that has been developed and validated as a rapid, reliable, and accurate method for simultaneous determination of ternary mixtures of BZs products diazepam, flunitrazepam, and lorazepam. A set of standards based on modified full factorial design was prepared to build the model. After validation with an external test set the model successfully predicted the concentration of three BZs despite accused overlapping of the voltammetric signals.

The work presented herein shows a promising tool for simultaneously determining mixtures of these regulated substances in a fast, cost efficient, and reliable manner with performance close to that of sophisticate equipment such as HPLC. The ET developed was able to quantify the individual components of ternary mixtures of diazepam, flunitrazepam, and lorazepam with a NRMSE as low as 0.105 for the independent sample subset.

However, the drawback of the reported procedure might be the need for a specifically defined buffer and measurement media, i.e., the Britton–Robinson buffer, although the involved effort for a given sample is minimum. Nevertheless, future works should consider compatibility with different biologic media, such as urine or blood serum. These conditions need to be further explored in order to maximize the strong points of the ET approach. The presented work nevertheless opens a promising forensic/illicit drug analysis as an alternative to traditional methodologies.

## Figures and Tables

**Figure 1 sensors-19-05002-f001:**
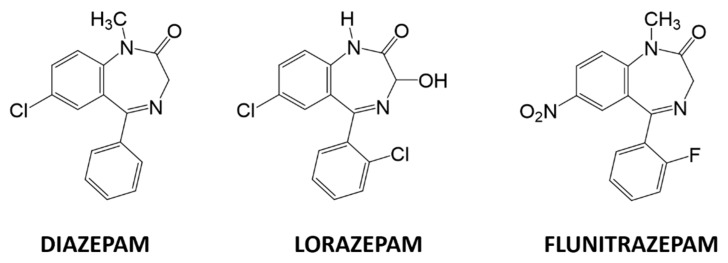
Chemical structures of diazepam, lorazepam, and flunitrazepam.

**Figure 2 sensors-19-05002-f002:**
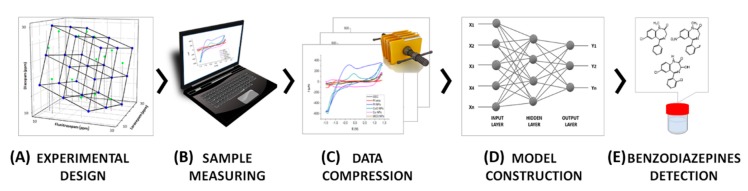
Scheme depicting the main steps of the ET approach used in the simultaneous determination of BZs.

**Figure 3 sensors-19-05002-f003:**
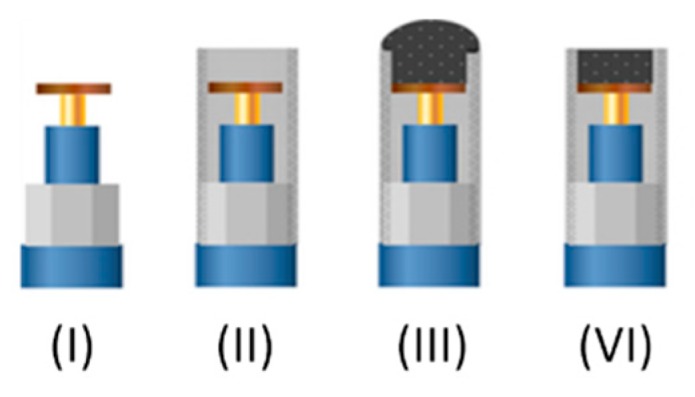
Scheme of the step-by-step construction of graphite-epoxy composite electrodes. (**I**) Copper disk soldered to the connector. (**II**) Assembly into the PVC tube. (**III**) Incorporation of graphite-epoxy mixture. (**VI**) Polishing of the hardened surface.

**Figure 4 sensors-19-05002-f004:**
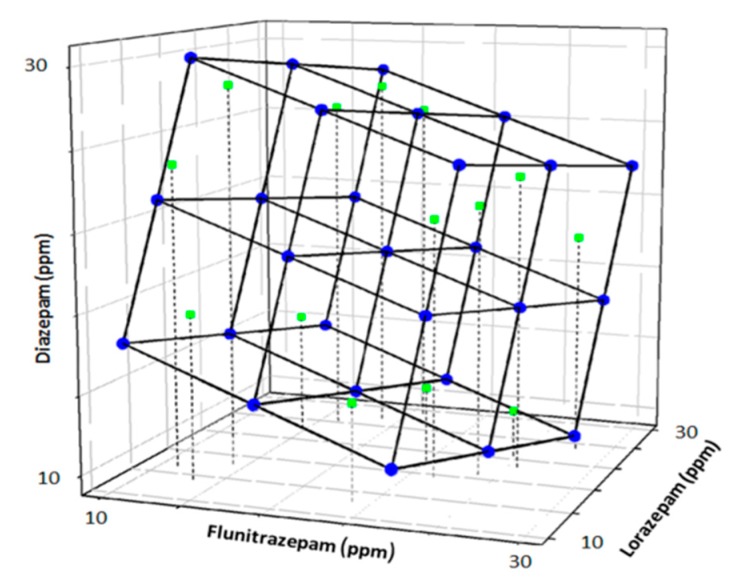
Scheme of the modified full factorial design used as train subset (blue) and randomly generated external test subset (green).

**Figure 5 sensors-19-05002-f005:**
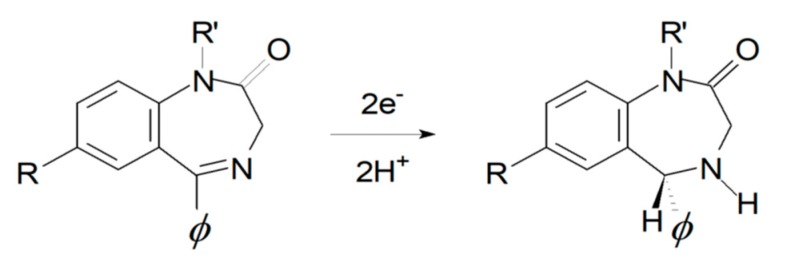
The oxidation of 4,5-azomethine redox mechanism.

**Figure 6 sensors-19-05002-f006:**
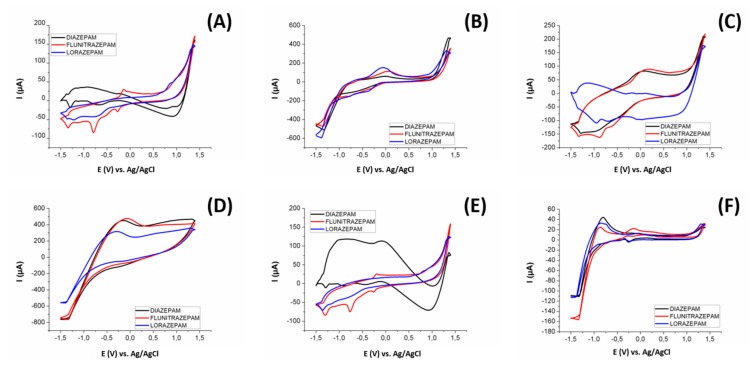
Voltammograms obtained with the three benzodiazepines used in this study (25 ppm) using a (**A**) bare epoxy-graphite composite electrode, and graphite electrodes modified with (**B**) CuO nanoparticles, (**C**) Cu nanoparticles, (**D**) Pt Nanoparticles, (**E**) WO_3_ nanoparticles, and a (**F**) Pt metal electrode.

**Figure 7 sensors-19-05002-f007:**
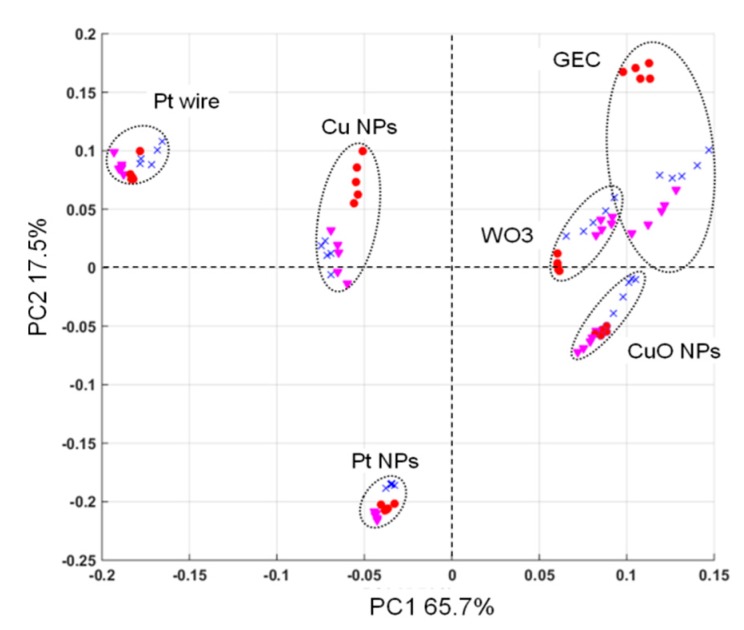
Score plot of the first two principal components obtained after PCA. Five replicates for each compound (concentration of 25 ppm) were measured with the six sensors in the ET array. Diazepam, (red circle); flunitrazepam (pink triangle), and lorazepam (blue cross).

**Figure 8 sensors-19-05002-f008:**
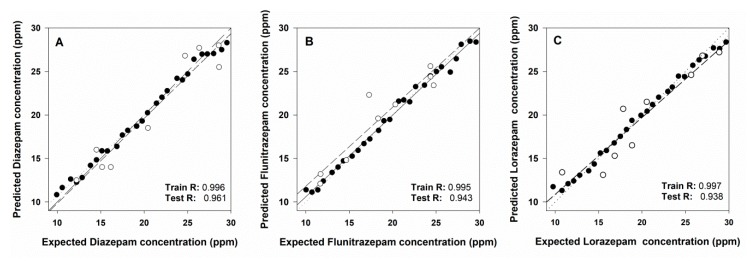
Obtained vs. expected concentrations results plots for the training set (black dots) and the testing set (white dots) for (**A**) diazepam, (**B**) flunitrazepam, and (**C**) lorazepam.

**Table 1 sensors-19-05002-t001:** Results of the fitted regression lines for obtained vs. expected values for the training and testing subsets of samples and the three considered compounds.

Set	Analyte	R^2^	Slope	Intercept (ppm)	NRMSE	Combined NRMSE
Training Set (n = 27)	Diazepam	0.996	0.932	1.358	0.035	
	Flunitrazepam	0.996	0.939	1.196	0.034	0.034
	Lorazepam	0.997	0.923	1.498	0.034	
Testing Set (n = 9)	Diazepam	0.961	0.977	0.075	0.105	
	Flunitrazepam	0.943	0.909	2.714	0.113	0.106
	Lorazepam	0.938	0.884	2.044	0.114	

**Table 2 sensors-19-05002-t002:** Analytical results of the work presented herein and most relevant references, This table compares the substances detected, the sensor and employed electrochemical technique, the linear range, and the limit of detection.

Substances.	Electrode	Technique	Lineal Range (M)	Limit of Detection (µM)	Reference
Diazepam	Ag nanodendrimers supported by graphene nanosheets modified glassy carbon electrode	Differential pulse voltammetry	1.0 × 10^−7^–1.0 × 10^−6^	0.086	[15]
Diazepam	Unmodified SPCE	Differential pulse adsorptive stripping voltammetry	2.5 × 10^−5^–1 × 10^−3^	6.3	[16]
Diazepam	Fullerene functionalized carbon nanotubes/ionic liquidnanocomposite GCE	Differential pulse voltammetry	0.3 × 10^−6^–7 × 10^−4^	0.0087	[17]
Diazepam	Multiwall Carbon Nanotube-Ionic Liquid Modified Paste Electrode	Square wave voltammetry	7 × 10^−8^–2.5 × 10^−6^	0.014	[23]
Lorazepam	Modified polypyrrole@sol-gel@gold nanoparticles/pencil graphite electrode	Cyclic voltammetry	0.2 × 10^−9^–2 × 10^−9^	0.0009	[18]
Flunitrazepam	SPGE	Cyclic voltammetry	3.2 × 10^−8^–6.4 × 10^−7^	0.019	[20]
Diazepam Flunitrazepam Lorazepam	Metal nanoparticle modified graphite epoxy sensor array	Cyclic voltammetry–Electronic tongue	3.5 × 10^−5^–1.1 × 10^−4^ 3.2 × 10^−5^–9.6 × 10^−4^ 3.1 × 10^−5^–9.3 × 10^−5^	6.0 5.6 4.6	This work

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
