# Peer review of "Voltammetric Electronic Tongue for the Simultaneous Determination of Three Benzodiazepines"

_sensors, 2019, doi:10.3390/s19225002_

Round 1

Reviewer 1 Report

Authors reports to use voltammetric sensors simultaneously detect Diazepam, Lorazepam and Flunitrazepam. It introduced an interesting algorithm for the detection. I would like to recommend its publication in Sensors after some minor revisions.

Comments:

In the manuscript, Authors used PCA to reduce dimension of data into 2. However, what is input of PCA? All points from CV curve or some special value? It’s better to give more details about neural network model’s input and output. It will help reader to understand easier how the network works. Authors said (DWT) was employed to reduce sample dimension. But. detail component or the approximation are kept after this pre-processing? Why are training set in figure.8 not same for predicted and expected concentrations. For training set, should these two be same?

Author Response

Authors reports to use voltammetric sensors simultaneously detect Diazepam, Lorazepam and Flunitrazepam. It introduced an interesting algorithm for the detection. I would like to recommend its publication in Sensors after some minor revisions.

Author’s comments: We thank the reviewer for the appreciation of the work carried out and the feedback. Following his advice, requested changes and raised questions have been addressed to improve the quality of the manuscript.

In the manuscript, Authors used PCA to reduce dimension of data into 2. However, what is input of PCA? All points from CV curve or some special value?

Author’s comments: The data input to PCA transformation are the individual voltammograms from each sensor (and each compound tested). Complete voltammograms are used in here. No feature extraction is performed for this examination/inspection of the data. A comment has been added in the text to clarify this point.

It’s better to give more details about neural network model’s input and output. It will help reader to understand easier how the network works.

Author’s comments: In Page 7, lines 237-238 there is the proper description of input/output details of the neural network model finally selected. The configuration had 144 input neurons, 5 hidden neurons, 3 output neurons, and transfer functions satlins in the hidden layer, and tansig in the output layer. This means the three benzodiazepines concentrations are predicted in a unique operation. Some extra info was added in this case, related to the pre-processing stage and the dimensions associated.  

Authors said (DWT) was employed to reduce sample dimension. But. detail component or the approximation are kept after this pre-processing?

Author’s comments: Exact transformation for compression suggests using approximation coefficients for next stage. This is in analogy to the Fourier transform, where normally low frequency coefficients contain the interesting information, and high frequency coefficients are more related to noise.  This specific detail has been added to the text.

Why are training set in figure.8 not same for predicted and expected concentrations. For training set, should these two be same?

Author’s comments:  Fig 8 displays the comparison of predicted concentrations vs. expected ones, for the three BZs and separated in train subset (black dots, 36 points) and external test subset (white circles, 9 points). Both subset follow more or less precisely the diagonal line (also in the graph), which indeed means predicted are equal to expected. In this sense, train subset normally fits much better the expected values, precisely because the model is obtained from these values. In fact, is only the proper behaviour for the external test subset that gives validity to the study, i.e. demonstrates there is no ‘overfitting’. In principle, the building of the response model looks that in both cases these are the same, only the really significant ones are those of the external test subset, as they are the ones showing the generalization ability of the developed model, i.e. the utility to predict new samples, different to the ones used for training.

Specific information to be deduced from Fig 8 is the larger dispersion observed for the test subset (Separation from the diagonal line), something that it is also habitual, as these samples, being completely external to the model building (they did not intervene at all in the developing of the model) get more difficulties in their correct prediction. According to referee’s suggestion, a specific comment on the fact has been added in Page 7.

In addition to the comments above, these facts are quantified numerically in the Table summarizing performance of the developed model (Table 1), where the reduced values of NRMSE exactly mean errors between predicted and expected are practically zero, or where the fitted comparison line is practically the diagonal line (y=x), i.e. predicted and expected values are equivalent. And the usual results are exactly as shown in there: for the training subset, equivalence is much closer that for the test subset and it is easily attained, whereas the main value is if the test subset also follows the expected trend.

Reviewer 2 Report

The authors present a voltammetric electronic tongue comprised of modified epoxy-graphite electrodes with oxide and metallic nanoparticles and metallic electrodes to simultaneously detect Diazepam, Lorazepam and Flunitrazepam ternary mixtures. Artificial neural network model was built up to predict the concentration of the three compounds. After some points have been clarified, the paper may be considered for publication.

Please carefully revise English grammar and typographical errors. e.g. page 2, l. 72.; page 3, l. 102; page 1, l. 37; page 5, l. 187; page 6, l. 202; caption of figure 6…

Authors discuss in the introduction that identification of BZs is interesting for criminology and forensic studies to detect them in blood, urine and hair samples. However, only standard samples were assessed in this study. Please clarify.

I also suggest informing in the Introduction what is the expected concentration of BZs to be found in blood, urine and hair samples. Because the reason to study these chemical inputs in a concentration of 10 to 30 ppm is not explained.

Please provide the concentration of Britton-Robinson buffer.

Please add more information about metallic and oxide nanoparticles and noble metals employed in this work to fabricate the electrodes. Inform also the area of counter electrode. They are not clearly described in experimental section.

Please explain how you concluded that 5% of modifier is enough to modify the electrode response. If necessary, add references.

Please clarify in the manuscript which voltammogram was employed for data analysis, first or second cycle.

I suggest changing the name of electrode modification, because it looks that were employed only nanoparticles, and not a composite.

Please add the compression value obtained to other levels. Is level 3 the best condition for data compression?

Please explain why satlins and tansig functions provided the best results. What problems were encountered with another 200 functions?

How many replicates were performed for each solution? Does the surface of electrodes renewed after each batch of experiment to test its reproducibility?

Please correct in Fig. 6 the legend of x-axis to E (V) vs. Ag/AgCl.

LOD values and a table comparing the obtained analytical parameters with ones reported in literature could be added to the text.

Author Response

The authors present a voltammetric electronic tongue comprised of modified epoxy-graphite electrodes with oxide and metallic nanoparticles and metallic electrodes to simultaneously detect Diazepam, Lorazepam and Flunitrazepam ternary mixtures. Artificial neural network model was built up to predict the concentration of the three compounds. After some points have been clarified, the paper may be considered for publication.

Authors’ comments. We thank the reviewer for the positive revision done, and the interesting observations placed. All his comments have been considered for final improvement of the manuscript.

Please carefully revise English grammar and typographical errors. e.g. page 2, l. 72.; page 3, l. 102; page 1, l. 37; page 5, l. 187; page 6, l. 202; caption of figure 6…

Author’s comments: English was revised in specific points commented, and also through the whole manuscript, with the general aim of reducing most of typos and help in readability.

Authors discuss in the introduction that identification of BZs is interesting for criminology and forensic studies to detect them in blood, urine and hair samples. However, only standard samples were assessed in this study. Please clarify.

Author’s comments:  This manuscript presents a simultaneous determination and quantification of three different benzodiazepines in Britton-Robinson buffer as a proof of concept. This simultaneous quantification is a novel work as recent articles only determine and quantify BZs individually. As a preliminary work in the determination of BZs the authors have focus on the simples matrix, that is buffer media samples. In future works the authors will explore different media including urine, plasma and serum/blood.

I also suggest informing in the Introduction what is the expected concentration of BZs to be found in blood, urine and hair samples. Because the reason to study these chemical inputs in a concentration of 10 to 30 ppm is not explained.

Author’s comments:  As commented in the previous response the work presented herein is a proof of concept. The range of concentration was chosen to ensure a clear voltammetric response of the different BZs in each of the selected sensors. In future works, the authors will fine-tune the concentration range employed in the model, as well as the limits of detection for each individual BZs, in order to employ the electronic tongue approach in more complex media such as blood or urine.

Please provide the concentration of Britton-Robinson buffer.

Author’s comments:  This specific info has been added to the manuscript, as demanded. Britton-Robinson multisaline buffer was prepared using 0.04 M H3BO3, 0.04 M H3PO4, 0.04 M CH3COOH and 0,1 M NaOH.

Please add more information about metallic and oxide nanoparticles and noble metals employed in this work to fabricate the electrodes. Inform also the area of counter electrode. They are not clearly described in experimental section.

Author’s comments:  Extra information concerning exact details of commercial nanocomponents used to modify used electrodes has been incorporated in the text, as suggested.  

Please explain how you concluded that 5% of modifier is enough to modify the electrode response. If necessary, add references.

Author’s comments:  Preliminary studies in the laboratory already demonstrated this amount of modifier is enough to provide differentiated signal. In this point, we must comment the goal is not to obtain maximum different signal possible, but stable and reproducible signal. Many previous research papers from our laboratory supports the correct operation of these experimental conditions.  Among others:

Talanta 179 (2018) 70–74

Talanta 162 (2017) 218–224

Electroanalysis 28 (2016) 1894–1900

Journal of Pharmaceutical and Biomedical Analysis 114 (2015) 321–329

Electrochimica Acta 120 (2014) 180– 186

Electroanalysis 25 (2014) 1635–1644

Sensors and Actuators B 177 (2013) 989– 996

Please clarify in the manuscript which voltammogram was employed for data analysis, first or second cycle.

Author’s comments:  Just the fourth voltammogram of a series of 4 is usually the one used for ET applications, with the aim of gaining in stability. A comment has been added in the text, as demanded.

I suggest changing the name of electrode modification, because it looks that were employed only nanoparticles, and not a composite.

Author’s comments:  The terminology related to the electrodes used has been revised to avoid confusion. We specially thank the reviewer for this clarification suggestion.

Please add the compression value obtained to other levels. Is level 3 the best condition for data compression?

Author’s comments:  Additional data related to the compression details, together with % of compression has been added in the text.  Additional compression levels presented lose of important information, degrading final performance of the data treatment.

Please explain why satlins and tansig functions provided the best results. What problems were encountered with another 200 functions?

Author’s comments:  Many configurations are possible when defining an artificial neural network (ANN) architecture, the only way to obtain the proper one is by systematic evaluation of all configurations possible. Prediction of the final option is normally not possible, although accumulated experience tells us that a linear function in conjunction with a non-linear one offers normally the best performance. For this purpose, in the study we evaluated the combinations between 2 and 9 neurons in the hidden layer, and the 5 transfer functions (hardlins, purelin, satlins, logsig and tansig) both in the hidden and output layers, constituting in this way the (8x5x5=200) configurations. Other details of the fitting is that we use normally Bayesian Regularization as the training algorithm, and that we provide a number of replicates during training, with random reinitialization of the ANN weights, for consistency of the final result. The configuration with best Normalized Root Mean Squared Error (NRMSE), which was the one better representing the training samples, measured through this parameter, was the one finally selected. The other 199 configurations then, were ANN models with poorer representation of the data.

How many replicates were performed for each solution? Does the surface of electrodes renewed after each batch of experiment to test its reproducibility?

Author’s comments:  Only one measurement experiment (in parallel obtaining the 6 voltammograms) was performed per sample. The expended time in this stage is one of the limitations of the approach, although it can be accelerated through the use of automatic analysis systems (FIA or SIA), thing we have done in the past.  In the experimental protocol, we incorporate an electrochemical cleaning stage, to help in stability and reproducibility of measurements. This is already explained in the manuscript.

Please correct in Fig. 6 the legend of x-axis to E (V) vs. Ag/AgCl.

Author’s comments:  Figure 6 legend has been amended, as demanded.

LOD values and a table comparing the obtained analytical parameters with ones reported in literature could be added to the text.

Author’s comments:  A comparison table has been elaborated, and added to the end of Results section, as Table 2. This table surely outlines the advantages of the developed approach in front of other voltammetric sensor variants appeared in the literature. We thank the reviewer for this useful suggestion.

Round 2

Reviewer 2 Report

The authors have improved the manuscript according to reviewers' suggestions and now it can be acceptable for publication.